# Heterogeneous Tumor-Immune Microenvironments between Primary and Metastatic Tumors in a Patient with *ALK* Rearrangement-Positive Large Cell Neuroendocrine Carcinoma

**DOI:** 10.3390/ijms21249705

**Published:** 2020-12-19

**Authors:** Takahiro Tashiro, Kosuke Imamura, Yusuke Tomita, Daisuke Tamanoi, Akira Takaki, Kazuaki Sugahara, Ryo Sato, Koichi Saruwatari, Shinya Sakata, Megumi Inaba, Sunao Ushijima, Naomi Hirata, Takuro Sakagami

**Affiliations:** 1Department of Respiratory Medicine, Kumamoto Chuo Hospital, Kumamoto-shi, Kumamoto 860-8556, Japan; tmaetashi19831210@yahoo.co.jp (T.T.); tamanoidaisuke@me.com (D.T.); akr.tkk0183@gmail.com (A.T.); sugachin_281279@yahoo.co.jp (K.S.); m-inaba@kumachu.gr.jp (M.I.); s-usijima@kumachu.gr.jp (S.U.); n-hirata@kumachu.gr.jp (N.H.); 2Department of Respiratory Medicine, Graduate School of Medical Sciences, Kumamoto University, Kumamoto-shi, Kumamoto 860-8556, Japan; imakou1013@gmail.com (K.I.); rmkqq751@ybb.ne.jp (K.S.); sakata-1027@hotmail.co.jp (S.S.); stakuro@kumamoto-u.ac.jp (T.S.); 3Laboratory of Stem Cell and Neuro-Vascular Biology, Genetics and Developmental Biology Center, National Heart, Lung, and Blood Institute, National Institutes of Health, Bethesda, MD 20814, USA; ryosato.1981@gmail.com

**Keywords:** anaplastic lymphoma kinase (ALK), B cell, heterogeneity, immune checkpoint, large cell neuroendocrine carcinoma (LCNEC), neuroendocrine tumors (NET), regulatory T-cells (Tregs), programmed death-ligand 1 (PD-L1), tumor-infiltrating lymphocyte (TIL), tumor-immune microenvironments (TIME)

## Abstract

Evolution of tumor-immune microenviroments (TIMEs) occurs during tumor growth and dissemination. Understanding inter-site tumor-immune heterogeneity is essential to harness the immune system for cancer therapy. While the development of immunotherapy against lung cancer with driver mutations and neuroendocrine tumors is ongoing, little is known about the TIME of large cell neuroendocrine carcinoma (LCNEC) or *anaplastic lymphoma kinase* (*ALK*) rearrangement-positive lung cancer. We present a case study of a 32-year-old female patient with *ALK*-rearrangement-positive LCNEC, who had multiple distant metastases including mediastinal lymph-node, bilateral breasts, multiple bones, liver and brain. Multiple biopsy samples obtained from primary lung and three metastatic tumors were analyzed by fluorescent multiplex immunohistochemistry. Tissue localizations of tumor-infiltrating lymphocytes in the tumor nest and surrounding stroma were evaluated. T cell and B cell infiltrations were decreased with distance from primary lung lesion. Although each tumor displayed a unique TIME, all tumors exhibited concomitant regression after treatment with an ALK-inhibitor. This study provides the first evidence of the coexistence of distinct TIME within a single individual with *ALK*-rearrangement-positive LCNEC. The present study contributes to our understanding of heterogeneous TIMEs between primary and metastatic lesions and provides new insights into the complex interplay between host-immunity and cancer cells in primary and metastatic lesions.

## 1. Introduction

Evolution of tumor-immune microenvironments (TIMEs) occurs during tumor growth and dissemination [1]. Accumulating evidence suggests that immune microenvironments have an influence on tumor initiation and response to cancer therapy [1,2,3]. It has been shown that the unique classes of TIME exist within a patient’s tumors [1,2]. A recent study suggested that multiple distinct TIMEs co-exist within a single individual with ovarian cancer and the heterogeneous TIMEs were associated with the heterogeneous fates of metastatic lesions clinically observed post-therapy [2]. Therefore, understanding the inter-site immune heterogeneity is critical and urgent challenges in development of treatment modalities to overcome intra-patient tumor heterogeneity [1,2,4,5,6].

The *ALK* fusion genes has been detected in approximately 2–5% of non-small cell lung cancer (NSCLC), most of which are adenocarcinoma [7,8]. Large cell neuroendocrine carcinoma (LCNEC) is a rare malignant tumor accounting for only 2 to 3% of all primary lung cancers and an aggressive malignancy with poor prognosis [9,10]. *ALK*-rearrangement is extremely rare event in LCNEC and only a few cases of *ALK*-rearrangement positive LCNEC have been reported [9,11,12]. Recent studies have shown that driver mutations including *anaplastic lymphoma kinase* (*ALK*) determine the composition of the TIME [13,14,15]. Understanding the complexity and diversity of the immune context of the tumor microenvironment is essential to harness the immune system against these cancers, however, little is known about the heterogeneity of TIME due to the rarity of *ALK*-rearrangement in LCNEC.

Immunotherapy with immune checkpoint blockade has demonstrated significant clinical activity across various cancer types [1,16,17]. Drug development of immunotherapy targeting NSCLC with a driver mutation involving *ALK* and advanced neuroendocrine tumors (NETs) including pulmonary LCNEC are currently ongoing [18,19]. However, clinical benefit from immunotherapy is restricted to a proportion of patients with NSCLC with driver mutations or NETs, highlighting the importance of investigating the heterogeneity of the TIME in these tumors [18,19,20].

Here, we present a case study of 32-year-old female patient with *ALK*-rearrangement positive LCNEC, who had multiple distant metastases, including mediastinal lymph node, right and left breasts, multiple bones, liver and brain. We investigated that multiple biopsy samples of the primary lung tumor, mediastinal lymph node, metastatic mammary and liver tumors by using fluorescent multiplex immunohistochemistry. The tissue localizations of tumor-infiltrating lymphocyte (TIL) subpopulations; CD3^+^ T cells, CD8^+^ T cells, CD3^+^FOXP3^+^ T cells (regulatory T cells, Tregs), and CD20^+^ B cells in tumor cell nest and surrounding stroma were profiled and quantified by automated quantitative pathology imaging system in primary and metastatic tumors.

This study provides the first evidence of the coexistence of distinct TIMEs within a single individual with LCNEC with an *ALK* rearrangement. CD3^+^ T cell and CD20^+^ B cell infiltrations were decreased with the distance from primary lung lesion. Each tumor lesion displayed a unique TIME, suggesting tailoring cancer therapy considering each TIME may be required to cure cancer. Primary lung tumor and metastatic lesions exhibited concomitant regression after treatment with ALK-inhibitor despite the heterogeneous TIME, although the tumors had eventually acquired resistance to ALK-TKI. The present study contributes to our understanding of the distinct TIMEs between primary and metastatic lesions.

## 2. Results

### 2.1. Case Presentation

A 32-year-old woman visited our hospital complaining of low back pain. She had a three pack-year history of cigarette smoking. Contrast-enhanced computed tomography (CT) revealed multiple osteolytic changes in the vertebral bodies, nodules in the left lower lung (Figure 1A), mediastinal lymph node (LN) enlargements (Figure 1B), bilateral breast tumors (Figure 1C), and multiple tumors in the liver (Figure 1D). The patient showed rapid progression of tumors and her performance status was two. Diffusion-weighted imaging of whole-body magnetic resonance imaging (MRI) revealed multiple abnormal signals in the rib, pelvis and vertebral bodies, which indicating multiple bone metastasis, in addition to the left intrapulmonary nodules, mediastinal lymph nodes and bilateral breast tumors (Figure 1E). Contrast-enhanced brain MRI revealed multiple, asymptomatic metastases in the brain (Figure 1F). Laboratory data demonstrated elevated tumor maker level of pro-gastrin-releasing peptide (ProGRP), 4362 pg/mL. The clinical stage was IVB (cT1bN3M1c).

Biopsy specimens from the lung, mediastinal LNs, both breast tumors, and liver revealed malignant cells organized into either solid nests or trabeculae of tumor cells with necrotic foci and rosette-like structures (Figure 2A). The immunohistochemistry (IHC) analyses showed that tumor cells were positive for thyroid transcription factor-1 (TTF-1) and neuroendocrine markers, including chromogranin A (Figure 2B), synaptophysin (Figure 2C), INSM1 (insulinoma-associated protein 1), and CD56 (Figure 2D). To differentiate from breast cancer, mammaglobin, estrogen receptor, progesterone receptor, and human epidermal growth factor receptor type 2 (HER-2) were investigated by IHC, but all were negative. These results indicated that all tumors were pathologically consistent with pulmonary LCNEC. Representative histological findings of the metastatic liver tumor are shown in Figure 2. ALK immunostaining was performed and showed diffuse positivity for all biopsied tissue samples (Figure 2E), and subsequent *ALK*-fluorescent in situ hybridization (FISH, break-apart assay) confirmed *ALK* rearrangement (Figure 2F). Other driver mutations were not detected.

Treatment with ALK-tyrosine kinase inhibitor (TKI), alectinib (600 mg/day) was administered. One month later, radiological evaluation revealed a rapid regression of all known lesions including metastatic brain tumors and the patient achieved a partial response (PR) to ALK-TKI therapy; however, contrast-enhanced brain MRI revealed marked increase in the size of the intracranial metastatic lesions and new brain metastases after 11 months of treatment with alectinib. A progressive disease was eventually evaluated according to Response Evaluation Criteria in Solid Tumor (RECIST) version 1.1. and whole-brain irradiation was performed.

### 2.2. Heterogeneous TIME between Primary and Metastatic Lesions in LCNEC with ALK Rearrangement

Understanding inter-site heterogeneity of the TIME is an urgent challenge in the development of treatment modalities to overcome intra-patient tumor heterogeneity and improve patients’ survival. However, inter-site heterogeneity of the TIME in LCNEC with *ALK* rearrangement remains to be uncovered. We investigated the immune contexture of pretreatment primary and metastatic tumors including mediastinal lymph node (LN), liver, right and left breast tumors by fluorescent multiplex immunohistochemistry. The fluorescent multiplex immunohistochemistry analysis has been shown to capture multidimensional data related to tissue architecture, spatial distribution of multiple cell phenotypes, and co-expression of signaling [21,22]. High-speed scanning of whole slide images was performed on stained tissue sections. Pan-cytokeratin of tumor cells and PD-L1 were simultaneously stained to evaluate the complex relationship among tissue architecture, spatial distribution of immune cells and expression of PD-L1, and then the tissue localizations of TIL were evaluated. CD3^+^ T cells (which include both CD4^+^ T cells and CD8^+^ T cells), CD8^+^ T cells, immunosuppressive CD3^+^FOXP3^+^ regulatory T cells (Tregs), and CD20^+^ B cells in the tumor nest and surrounding stroma were profiled and quantified by an automated quantitative pathology imaging system (Figure 3).

The quantitative analysis of fluorescent multiplex immunohistochemistry staining with high-speed scanning of whole slide images revealed that coexistence of distinct TIMEs within a single patient with *ALK* rearrangement-positive LCNEC. Each tumor displayed a unique TIME (Figure 4).

Interestingly, CD3^+^ T cell and CD20^+^ B cell infiltrations in tumors were decreased with the distance from primary lung lesion (Figure 4 and Figure 5).

It has been shown that the distribution and ability of lymphocytes to localize within tumor islets influence the outcome of tumor development and the responsiveness to cancer immunotherapy [23,24]. Thus, we further investigated the detail of tissue localizations of TIL subpopulations; CD3^+^ T cells, CD8^+^ T cells, Tregs, and CD20^+^ B cells in the tumor cell nest and surrounding stroma and quantified them by using automated quantitative pathology imaging system in primary and metastatic tumors. Although tumor PD-L1 expression was not detected in both primary pulmonary and metastatic LCNEC with *ALK* rearrangement, primary lung tumors had highest amount of CD3^+^ T cell, CD8^+^ T cell and CD20^+^ B cell infiltrations (Figure 4, Figure 5 and Figure 6). Higher CD8^+^ T cell infiltrations were detected in primary lung and breast tumors compared to mediastinal LN and liver tumors, however, which were mostly restrained in the stroma of tumors (Figure 4 and Figure 6). Treg infiltrations were observed in primary lung, mediastinal LN and breast tumors, but not detected in metastatic liver tumors (Figure 6). Higher Treg infiltrations into tumor nests were detected in mediastinal LN and breast tumors compared to primary lung tumors (Figure 6). Higher CD20^+^ B cell infiltrations were detected in primary lung and mediastinal LN tumors compared to breast and liver tumors, however, which were mostly restrained in the stroma of tumors (Figure 6). Only a few intratumoral CD3^+^ T cell, CD8^+^ T cell, Treg, and CD20^+^ B cell infiltrations were observed in metastatic liver tumors (Figure 6).

## 3. Discussion

Reciprocal interactions between tumor cells and infiltrating immune cells evolve as the tumor grows in progressing cancers [1,3]. Understanding heterogeneous TIMEs in *ALK* rearrangement-positive LCNEC may offer crucial information to optimize the use of immunotherapy, radiotherapy and chemotherapy and may be instructive to discover novel therapeutic approaches for improving the actual cure rate of advanced lung cancer [1]. Jiménez-Sánchez et al. have shown that multiple distinct TIMEs co-exist within a single individual and the heterogeneous TIME was associated with the heterogeneous fates of metastatic lesions, suggesting that understanding the diversity of the immune context of the tumor microenvironment is essential to harness the immune system against cancers [2]. *ALK* rearrangement is extremely rare event in LCNEC and only a few cases of *ALK* rearrangement positive LCNEC have been reported [11,12]. In addition, little is known about the TIME of LCNEC with *ALK* rearrangement, or indeed, the TIME of NET [11,15,19]. In the current study, we presented an exceptional case of a young patient with *ALK* rearrangement-positive LCNEC treated with ALK-TKI, who exhibited concomitant regression of both primary tumor and multiple metastatic lesions after the treatment despite the heterogeneous TIME. The present case is the youngest patient with *ALK* rearrangement-positive LCNEC reported to date [11]. We have confirmed *ALK* rearrangement positivity in the primary lung tumor and all metastatic sites except for metastatic bone and brain tumors. Although *ALK* rearrangement is uncommon in LCNEC, given the patients with *ALK* rearrangement-positive LCNEC may obtain increased benefit from ALK inhibitors than conventional chemotherapies, clinicians should routinely test for *ALK* rearrangements in patients with LCNEC, especially in young patients.

*ALK* rearrangement is most commonly seen in lung adenocarcinoma [7,25]. ALK-TKIs have dramatically improved overall response rates in NSCLC with *ALK* rearrangement in clinical trials relative to conventional chemotherapy [26,27]. However, patients eventually develop disease progression because of acquired resistance through various mechanisms, resulting in treatment failure [25,26]. Additionally, patients with brain metastasis of *ALK* rearrangement-positive NSCLC have been known to have shorter progression-free survival (PFS). Pulmonary LCNEC is a rare tumor accounting for only 2 to 3% of all primary lung cancers and carries an aggressive clinical behavior and poor prognosis [9,10,28]. Based on retrospective analyses, a poor 5-year overall survival was indicated with a high incidence of recurrence after surgery, even in stage I disease [9,10]. A recent study developed an elaborate nomogram to estimate individualized prognosis for LCNEC in terms of 3-year and 5-year OS [10]. According to the prediction model for LCNEC prognosis, the estimated 3-year survival for the current patient was less than 10%. Although there is no validated therapeutic approach for advanced pulmonary LCNEC due to the rarity and lack of clinical trials, patients with advanced LCNEC generally receive platinum-based doublet chemotherapy as first line treatment, however, their prognosis is poor [9,10,19]. Thus, new agents including immunotherapy are under clinical investigation to improve LCNEC patients’ outcome [10,19]. The available evidence is currently insufficient to tailor an optimal treatment strategy specified for patients with pulmonary LCNEC.

*ALK* rearrangement-positive tumors are at high risk of relapse and patients with *ALK* rearrangement-positive tumors usually develop disease progression because of acquired resistance [26]. In addition, LCNEC has a generally poor prognosis similar to small-cell lung cancer [10]. Thus, understanding the inter-site immune heterogeneity of these tumors is critical and urgent challenges in the development of treatment modalities to overcome intra-patient tumor heterogeneity exist. Although the current patient with *ALK* rearrangement-positive LCNEC received ALK-TKI, alectinib and achieved a PR for 11 months, the brain tumor had acquired resistance to ALK-TKI similar to a case previously reported [12].

Understanding the complex TIME offers the opportunity to make better prognostic evaluations and select optimum treatments [1,2,24]. Accumulating evidence suggests that a high density of tumor-infiltrating CD8^+^ T cells strongly associates with improved clinical outcomes [1,5,14,24,29]. A recent study has shown that tumor cell nest enriched in CD8^+^ T cell infiltration reflects the ability of CD8^+^ T cells to infiltrate tumor cell nests, which is independently associated with better overall survival of lung cancer patients [23]. Tregs have immunosuppressive activity and play a critical role in negatively regulating anti-tumor immune responses [30,31,32]. A high density of tumor-infiltrating CD20^+^ B cells has been shown to correlate with prolonged survival in patients with a wide variety of human cancers including lung cancer [33,34]. Therefore, we have analyzed paraffin-embedded tissue sections to obtain insight into the localization of these TILs relative to tumor nest (stained by pan-cytokeratin) and the surrounding tumor stroma. Using fluorescent multiplex immunohistochemistry and quantitative pathology imaging analyses, we found that primary lung tumors and multiple metastatic tumors have heterogeneous TIMEs, which suggests that an optimal therapeutic approach tailored to each metastatic site considering its heterogeneous TIME may be required to cure advanced cancer.

In a recent study, Muller P and colleagues investigated the heterogeneity of immune cell infiltrates between primary NSCLC and corresponding metastases [35]. In this study, primary tumors and corresponding metastases from 34 NSCLC patients were extensively analyzed by immunohistochemistry for CD4, CD8, CD11c, CD68, CD163 and PD-L1. Interestingly, this study reported that the CD8/CD4 ratio and CD8/CD68 ratio were significantly reduced in metastatic tumors compared with the corresponding primary tumors, suggesting a tolerogenic and tumor-promoting microenvironment at the metastatic site. Interestingly, we found that CD3^+^ T cell and CD20^+^ B cell infiltrations in tumors were decreased with the distance from primary lung lesion, implying that the distance from primary lesion might impact on TIME. Our results demonstrated here may help to explain the heterogeneous tumor responses to the therapy in primary tumor and metastatic lesions, which are often observed in advanced cancer patients.

Although tumor PD-L1 expression was not detected in both primary and metastatic tumors, primary lung tumors had highest amount of CD3^+^ T cell, CD8^+^ T cell and CD20^+^ B cell infiltrations compared with metastatic tumors. It has been shown that anti-PD-1/PD-L1 inhibitor monotherapies or in combination with platinum-based chemotherapies improved patients’ survival in metastatic NSCLC regardless of PD-L1 expression in the tumor [36,37,38,39]. These results suggest that the primary lung tumor in our case may mount a protective immune response to attack tumor cells and have the potential to respond to immune checkpoint inhibitor monotherapy or combination therapy with an immune checkpoint inhibitor and chemotherapies despite the negative expression of PD-L1 in tumor cells [36,38,39,40].

CD8^+^ T cell infiltrations were very few in both tumor nests and stroma of mediastinal LN tumors although the highest Treg infiltration was observed in tumor nest among primary and metastatic tumors. It has been known that immunosuppressive Tregs express cytotoxic T-lymphocyte-associated protein 4 (CTLA-4) at higher levels and a combination of programmed cell death 1 (PD-1) and CTLA-4 blockades increases tumor-infiltrating effector T cell and reduces tumor-infiltrating Tregs [41,42]. These results imply that combined immunotherapy targeting the immune checkpoint receptors CTLA-4 and PD-1 may be effective for mediastinal LNs [19,43].

Liver metastatic patients with NSCLC that had been treated with anti-PD-1 antibody monotherapy were associated with reduced responses and progression-free survival, and the liver metastases were associated with reduced marginal CD8^+^ T cell infiltration [29]. In the current study, metastatic liver tumors had only a few intratumoral CD8^+^ T cell, and no Treg and CD20^+^ B cell infiltrations, which indicates the TIME of metastatic liver tumors is close to “immunological ignorance” type (TIL^−^, PD-L1^−^), an immunological state in which adaptive immunity is unable to recognize or respond to tumor cells [6,44]. These results suggest that a combination therapy with chemotherapies and immune checkpoint inhibitors or radiotherapy may be required for treating liver tumors [1,37,38,45].

The present study has notable limitations. Only one patient is involved in this study, thus further studies are needed to determine whether the principles discovered here apply to other lung cancer patients. Furthermore, myeloid cell subsets such as myeloid-derived suppressor cells, dendritic cells, tumor-associated macrophages (TAMs), and tumor-associated neutrophils in the TIME were not investigated although myeloid cells in tumors are considered to play a crucial role in the control of tumor growth and progression [46,47,48]. TAMs inhibit immuno-stimulatory signals and are implicated in the initiation and progression of the tumor, through the secretion of signaling molecules, such as vascular endothelial growth factor (VEGF), transforming growth factor beta (TGF-β), macrophage colony-stimulating factor (M-CSF), interleukins or chemokines (IL-10, IL-6, and CXCL-8) [49]. Factors secreted by TAMs, such as TGF-β, VEGF, CCL8, COX-2, MMP9, and MMP2 also contribute to the metastatic properties of cancer cells. In addition, TAMs are responsible for resistance to conventional antitumor treatments, such as chemotherapy, radiotherapy, or immune checkpoint inhibitors. However, we did not investigate TAMs in TIMEs of primary and metastatic tumors, which is a limitation. Cancer therapy including ALK inhibitors may change the TIME in primary and metastatic tumors. In current study we did not address the impact of ALK-TKI on TIME. Thus, further studies evaluating TIME pre- and post-therapy are needed to understand the complex interplay among cancer cells, host-immunity, and cancer therapy. Whole-exome sequencing and RNA-sequencing are important approaches to dissect the heterogeneity of complex biological systems [50,51,52,53]. Analyses of primary and metastatic tumors by whole-exome sequencing and RNA-sequencing pre- and post-therapy could provide crucial information to understand the complex interplay among cancer cells, host-immunity, and therapy. However, we could not use these analyses in the current patient. Circulating tumor cells (CTCs) are cancer cells that are shed from the primary or metastatic tumors into the bloodstream [54,55]. The enumeration and characterization of CTCs provided a minimally invasive and, therefore, repeatable, method despite being present in extremely low numbers, enabling the sampling of tumor cells from peripheral blood and monitoring PD-L1 expression on tumor cells over time. In addition, EML4-ALK rearrangements have been reported to be found in CTCs [56]. Thus, combining CTC and peripheral immune subset analyses may make it possible to longitudinally evaluate serial human specimens during treatment (at pre-treatment, early-on-treatment, and progression time points), which allow for deep analysis to unveil potential mechanisms of therapeutic resistance [16,50,57,58]. These approaches may contribute to develop treatment modalities to overcome intra-patient tumor heterogeneity.

In conclusion, this study provides the first evidence of the coexistence of distinct TIMEs within a single patient with *ALK* rearrangement-positive LCNEC. CD3^+^ T cell and CD20^+^ B cell infiltrations in tumors were decreased with the distance from primary lung lesion. Although tumor PD-L1 expression was not detected in primary and metastatic LCNEC with *ALK* rearrangement, primary lung tumors had highest amount of CD3^+^ T cell, CD8^+^ T cell and CD20^+^ B cell infiltrations. These results suggest that tailoring cancer therapy considering the TIME of each tumor lesion may be needed. The patient exhibited concomitant regression of both primary tumor and multiple metastatic lesions after treatment with ALK-inhibitor despite the heterogeneous TIME. Given the patients with *ALK* rearrangement-positive LCNEC may obtain increased benefit from ALK inhibitors than conventional chemotherapies, clinicians should routinely test for *ALK* rearrangement in patients with LCNEC. Although further studies are needed to determine whether the principals discovered here apply to other cancer patients, the present study may contribute to our understanding of the TIME of lung cancer and provide new insights into the complex interplay between host-immunity and cancer cells.

## 4. Materials and Methods

### 4.1. Patient

Pre-treatment tissue samples of a 32-year-old female patient with ALK-rearrangement positive LCNEC were analyzed. Multiple biopsy samples from primary lung tumor and three metastatic sites were analyzed by fluorescent multiplex immunohistochemistry. The Kumamoto Chuo Hospital Review Board approved the study (Approval Date, 29 June 2020). This study was approved by the patient and we obtained written consent form the patient. This report was prepared in accordance with the Helsinki Declaration.

### 4.2. Fluorescent Multiplex Immunohistochemistry

Formalin-fixed paraffin-embedded (FFPE) sections of primary lung tumor and three metastatic tumors (mediastinal lymph node, breast and liver) were analyzed. Paraffin-embedded tissue sections were de-waxed with xylene and rehydrated by gradient ethanol solution. They were then processed for antigen retrieval by 10 mM citrate antigen buffer (pH 6.0) via microwave radiation except for PD-L1. For PD-L1 staining, the tissue sections were processed by pH 9.0 citrate buffer via autoclave. The sections were incubated with 3% H_2_O_2_ for 5 min to inhibit endogenous peroxidase activity, washed with 0.05% Tween in TBS (TBST), exposed to blocking buffer (5% goat serum, 0.5% bovine serum albumin in PBS) for 20 min, and incubated for 60 min with primary antibodies. Fluorescent multiplex immunohistochemistry was performed with OPAL Multiplex Fluorescent Immunohistochemistry Reagents (PerkinElmer, Waltham, MA, USA) following the manufacturer’s protocol. As outlined in the Table 1, formalin-fixed paraffin-embedded (FFPE) sections of primary and metastatic tumors were stained by three sequences of primary antibodies; PD-L1 (Cell Signaling Technology, Danvers, MA, USA), pan-Cytokeratin (Abcam, Cambridge, United Kingdom) and CD8 (Nichirei, Tokyo, Japan), or pan-Cytokeratin, FOXP3 (Abcam, Cambridge, United Kingdom) and CD3 (Abcam, Cambridge, United Kingdom), or pan-Cytokeratin and CD20 (Abcam, Cambridge, United Kingdom). They were then washed with TBST, incubated with anti-mouse or anti-rabbit HRP polymer conjugated secondary antibodies (Nichirei, Tokyo, Japan) for 10 min except for PD-L1 which was incubated for 30 min, and washed again, after which immune complexes were detected with OPAL reagents (PerkinElmer, Waltham, MA, USA). Nuclei were counterstained with 40,6-diamidino-2-phenylindole dihydrochloride (DAPI) (DOJINDO, Kumamoto, Japan) in PBS, and whole sections were mounted in ProLong Diamond (Thermo Fisher Scientific, Waltham, MA, USA). Multiplex slides were observed with a fluorescence microscope (BZ-X700, Keyence, Osaka, Japan). Then, tissue localizations of tumor-infiltrating lymphocytes in the tumor cell nest and surrounding stroma were evaluated.

### 4.3. Quantitative Analysis of Fluorescent Multiplex Immunohistochemistry Staining

Four subsets of tumor-infiltrating lymphocytes (TILs) were identified: CD3^+^ CD8^+^ Nuclei^+^ (representing cytotoxic T cells), CD3^+^ FOXP3^+^ Nuclei^+^ (representing regulatory T cells [Tregs]), CD3^+^ Nuclei^+^ (representing T cells), and CD20^+^ Nuclei^+^ (representing B cells) immune cells. Cytokeratin of tumor cells and PD-L1 were simultaneously stained to evaluate the complex relationship among tissue architecture, spatial distribution of immune cells and expression of PD-L1. High-speed scanning of whole slide images was performed on stained tissue sections using 20× objective (BZ-X700.Keyence. Osaka, Japan). For comparison of quantitative marker expression, images were analyzed with Strata Quest (Tissue Genetics, Vienna, Austria). These TIL subsets were separately counted in the tumor nest (epithelial compartment) and surrounding stroma under high-power fields. For nuclei and tumor detection, size and staining intensity of DAPI and pan-Cytokeratin were adjusted. Then, cutoff thresholds of CD3, CD8, CD20 and FOXP3 were determined by two independent observers. Finally, all images were analyzed and statistics including number of each subset cells was generated automatically. As for Tregs, cells express both CD3 and FOXP3 were recognized as Tregs. Cells were classified as tumor nest or tumor stroma according to the relationship with pan-Cytokeratin-positive tumor cells.

## Figures and Tables

**Figure 1 ijms-21-09705-f001:**
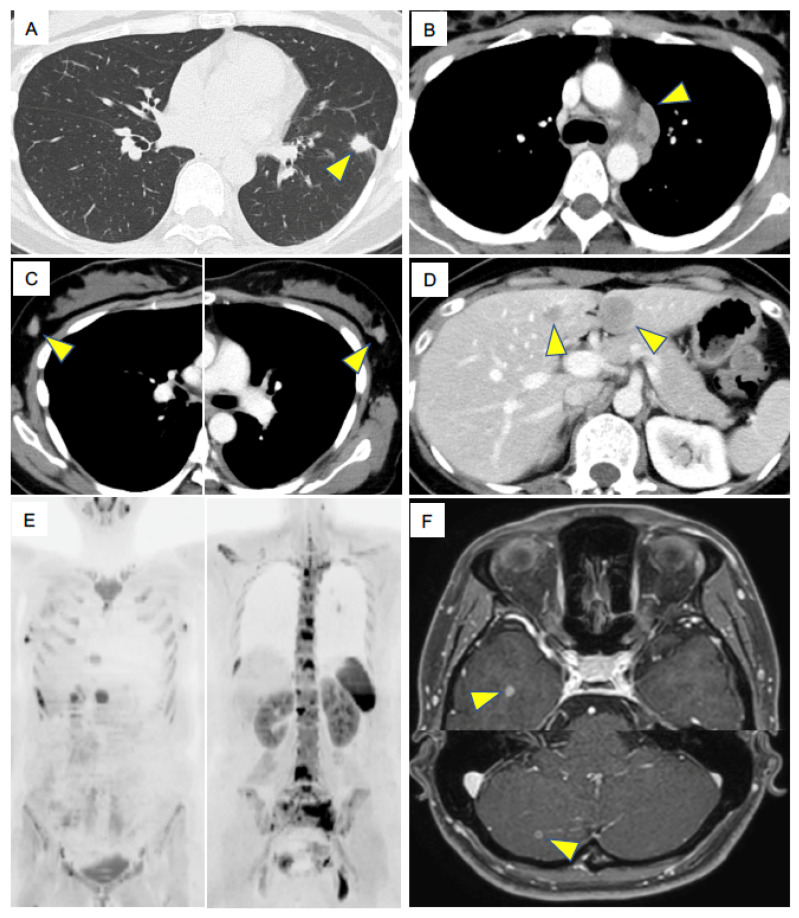
Key imaging results at diagnosis. (**A**). Chest computed tomography (CT) shows a primary lung tumor in the left lower lung. (**B**). Contrast-enhanced CT shows enlarged mediastinal lymph nodes (LNs). (**C**). Contrast-enhanced CT shows enlarged bilateral breast tumors. (**D**). Contrast-enhanced CT shows metastatic liver tumors. (**E**). Diffusion-weighted imaging of whole-body magnetic resonance imaging at the diagnosis shows multiple abnormal signals in the rib, pelvis and vertebral bodies, indicating multiple bone metastasis. Left panel also shows abnormal signals in bilateral breast tumors. (**F**). Contrast-enhanced brain MRI shows metastatic brain tumors. Arrowheads indicate metastatic tumors.

**Figure 2 ijms-21-09705-f002:**
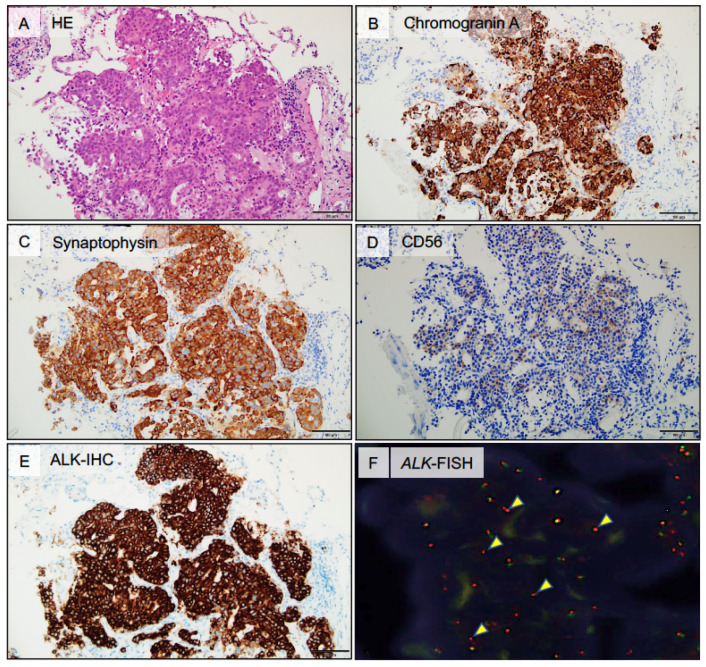
Histopathological findings of the mediastinal LN. (**A**), A hematoxylin and eosin–stained section from the liver tumor showing cells organized in solid nests or forming trabeculae with foci of necrosis and rosette-like structures. (**B**), Immunohistochemical analysis of the liver tumor showing strong diffuse chromogranin A positivity. (**C**), Immunohistochemical analysis of the liver tumor showing strong diffuse synaptophysin positivity. (**D**), Immunohistochemical analysis of the liver tumor showing CD56 positivity. (**E**), Immunohistochemical analysis showing strong diffuse ALK positivity (rabbit monoclonal antibody; D5F3). (**F**), Fluorescence in situ hybridization (FISH) analysis of the *ALK* locus using a break-apart probe strategy. The upstream and downstream of *ALK* gene were labeled red and green, respectively. Approximately 58% of tumor cells showed rearrangement at the *ALK* locus, as demonstrated by split red/green signals. Arrowheads indicate split pattern signals. ALK, anaplastic lymphoma kinase. Scale bars, 100 µm, are shown in panels.

**Figure 3 ijms-21-09705-f003:**
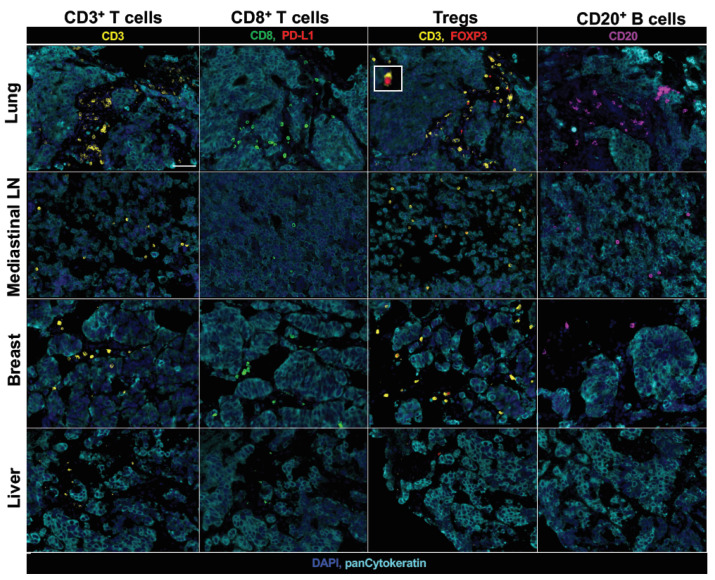
Multiplex fluorescent immunohistochemistry results. Representative images of lung, mediastinal LN, breast and liver tumors are shown. Formalin-fixed paraffin-embedded sections of each tumor were stained with one of three sequences of primary antibodies, CD3 (yellow), CD8 (green) and PD-L1 (red), CD3 and FOXP3 (red), or CD20 (purple). All sequences were also included with pan-Cytokeratin (light blue) for staining tumor cells, and nuclei were counterstained with DAPI (blue). PD-L1 expression was not identified in all tumors. The inserted panel in left upper of a representative lung tumor Treg image shows a Treg cell at high magnification. Scale bars, 50 µm, are shown in CD3^+^ T cells of lung tumor panel.

**Figure 4 ijms-21-09705-f004:**
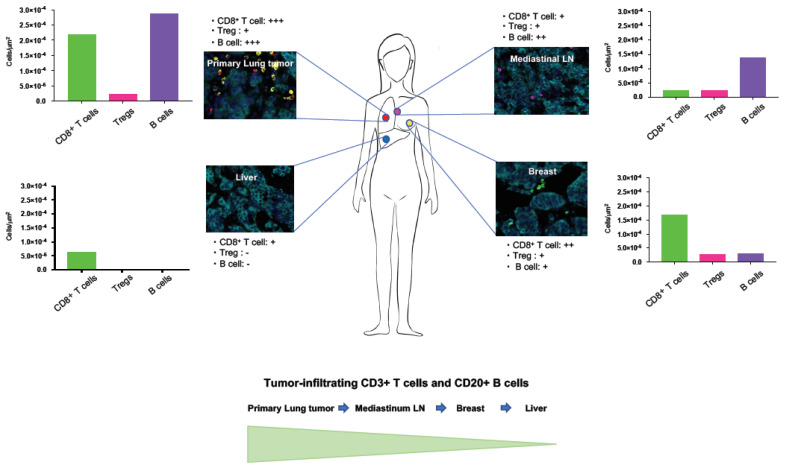
Heterogeneous tumor-immune microenvironments (TIMEs) in LCNEC with *ALK* rearrangement. Overall schema of TIMEs in primary and metastatic LCNEC with *ALK* rearrangement is shown. Biopsied lesions and representative images of lung, mediastinal LN, breast and liver tumors are shown. Tumor-infiltrating lymphocyte (TIL) subsets, CD3^+^ T cells (which include both CD4^+^ T cells and CD8^+^ T cells), CD8^+^ T cells, Tregs, and CD20^+^ B cells, were counted in primary and metastatic tumors. High-speed scanning of whole slide images was performed on stained tissue sections. Images of full sections were acquired and analyzed with automated quantitative pathology imaging system. All images were analyzed and statistics of the number of each types of cells were generated automatically. CD3^+^ T cell and CD20^+^ B cell infiltrations in tumors were decreased with the distance from primary lung lesion.

**Figure 5 ijms-21-09705-f005:**
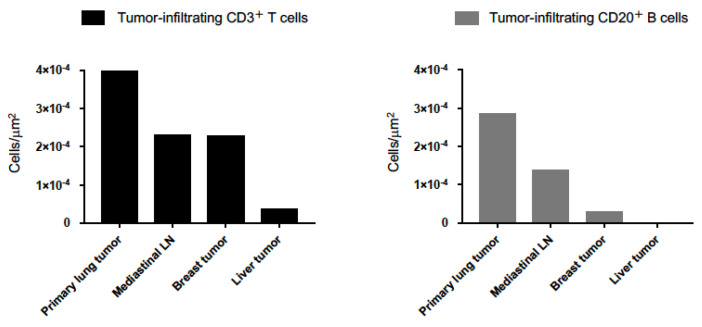
Quantification analysis of tumor-infiltrating CD3^+^ T cells and CD20^+^ B cells. High-speed scanning of whole slide images was performed on stained tissue sections. Images of full sections were acquired and analyzed with automated quantitative pathology imaging system. All images were analyzed and statistics of the number of tumor-infiltrating CD3^+^ T cells and CD20^+^ B cells was generated automatically. The number of tumor-infiltrating CD3^+^ T cells and CD20^+^ B cells was decreased with the distance from primary lung lesion.

**Figure 6 ijms-21-09705-f006:**
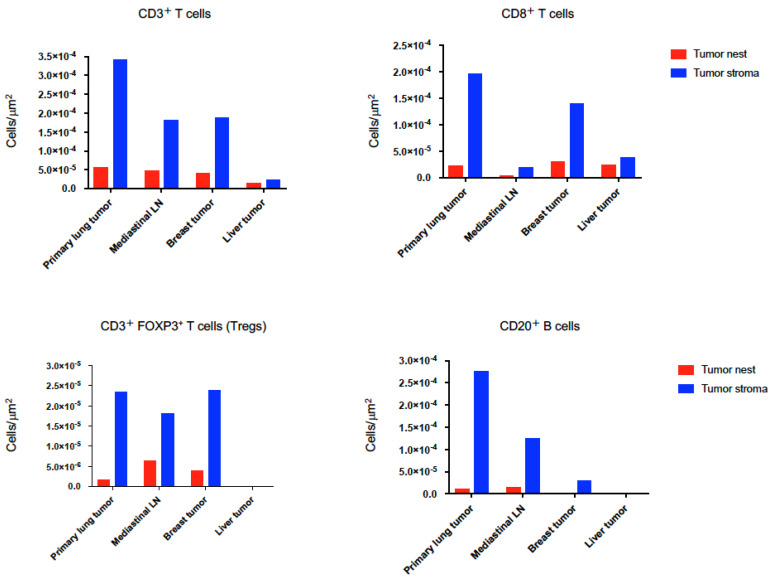
Quantification of CD3^+^ T cells, CD8^+^ T cells, CD3^+^ FOXP3^+^ T cells (Tregs), and CD20^+^ B cells. High-speed scanning of whole slide images was performed on stained tissue sections. Images of full sections were acquired and analyzed with automated quantitative pathology imaging system. All images were analyzed and statistics of the number of each types of cells were generated automatically. These tumor-infiltrating lymphocyte (TIL) subsets were separately counted in the tumor cell nest (epithelial compartment) and surrounding stroma under high-power fields. Cells were classified as tumor nest (red bars) or tumor stroma (blue bars) according to the relationship with pan-Cytokeratin-positive tumor cells.

**Table 1 ijms-21-09705-t001:** The list of antibodies used for fluorescent multiplex immunohistochemistry analysis.

Immune Subset	Antibody	Clone (Host)	Dilution	TSA Dyes
CD8^+^ T cells	CD8	C8/144B (mouse)	undiluted	520
	pan-Cytokeratin	AE1/AE3 + 5D3 (mouse)	1:200	570
	PD-L1	E1L3N (rabbit)	1:100	650
CD20^+^ B cells	CD20	L26 (mouse)	1:50	520
	pan-Cytokeratin	AE1/AE3 + 5D3 (mouse)	1:200	570
CD3^+^ T cells CD3^+^ FOXP3^+^ T cells	CD3	SP7 (rabbit)	1:100	520
	FOXP3	236A/E7 (mouse)	1:100	570
	pan-Cytokeratin	AE1/AE3 + 5D3 (mouse)	1:200	650

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
