# Peer review of "Heterogeneous Tumor-Immune Microenvironments between Primary and Metastatic Tumors in a Patient with ALK Rearrangement-Positive Large Cell Neuroendocrine Carcinoma"

_ijms, 2020, doi:10.3390/ijms21249705_

Round 1

Reviewer 1 Report

In this case study, the authors investigated the coexistence of the distinct evolution of tumor-immune microenvironment (TIME) between primary and multiple distant metastasis in 32-year-old female patient with ALK-rearrangement positive LCNEC. For the first time, they found that the primary lung tumor and metastatic lesions exhibited concomitant regression after treatment with ALK-inhibitor despite the heterogeneous TIME (CD3+ T cell and CD20+ B cell infiltrations decreased in relation to the distance from primary), although the tumors had eventually acquired resistance to ALK-TKI. This could provide new insights into the interplay between host-immunity and cancer cells among primary and metastatic lesions.

To my eyes, it is a good paper looking at the novelty of the aim, the methodology used for data analysis and the results are very interesting. The scientific content is quite good but the English style and language should be revised (there are few grammatical mistakes and words require some corrections). Moreover, I appreciate the scientific efforts to organize this paper and I think that the rationale is well-argued and the design of experiments is great. Secondly, the results are intriguing and organized in each section. The resolution of the figure 3F (Fluorescence in situ hybridization analysis of the ALK locus) needs to be improved. You might specify by arrows the exact point in which the ALK rearrangement occurs and the partner of ALK fusion (EML4 or others?). Likewise in figure 4. The reference list meets the quality requirements of our journal. Moreover, the results were clearly discussed and corroborated with what is shown. Finally, the data analyses were interpreted in a comprehensible manner. Overall, I agree with the authors in terms of methodologies, results and comments of data, supporting the proposal of their good clinical case presentation. I suggest to the authors to resume the materials and methods, especially the Fluorescent multiplex immunohistochemistry section.

In my opinion, this original article should be considered with minor revisions in light of comments/suggestions as indicated below.
As in any case study, the present study has important limitations.

1) It involves only one patient, and thus further studies are needed to determine whether the principles discovered here apply to other patients.

2) The interplay between treatment, somatic mutations, the immune system, and heterogeneous fates of the tumors cannot be untangled in this clinical case. For example, it is feasible that the ALK inhibitors treatment for this patient contributed to shaping the somatic mutations and the microenvironment of the tumors, but due to the descriptive nature of the study it was not possible to explore it. Please discuss this point.

3) Moreover, whole-exome sequencing and RNA-sequencing would have been important to deeply investigate the genetic, molecular, and cellular components that potentially underlie this differential growth. For example, I wonder if exists any evidence of gene-expression or somatic-alteration patterns (mutations, DNA amplification, and deep deletion) that differed between progressing (primary and distant metastases including mediastinal lymph-node, bilateral breasts, multiple bones, liver and brain) and regressing/stable tumors. Please justify this issue and explain why you didn’t perform NGS sequencing applied to this clinical case.

4) Tumors that are genetically more heterogeneous have less immune infiltrates (Senbabaoğlu et al., Genome Biol. 2016) and less benefit from checkpoint-blockade immunotherapies (McGranahan et al., Science. 2016). Thus, the exploration of the tumor-immune microenvironments (TIME) as a prognostic, diagnostic or predictive biomarker is an area of active research. In fact, TMB was predictive of clinical efficacy independent of PD-L1 expression, suggesting that both TMB and PD-L1 can be used as biomarkers for patient selection (Hellmann MD et al. Cancer Cell 2018). Recently, the isolation and culture of circulating tumor cells (CTCs) may be an alternative to tumor biopsies for the diagnosis of ALK rearrangement. Additionally, the effectiveness of ALK detection by FISH using cultured CTCs could be an interesting point to prove. What's your opinion about this? Please brief reply in the discussion.

5) It is known as endothelial cells respond to vascular endothelial growth factor (VEGF) from various sources including tumour cells, natural killer (NK) cells and tumour-associated macrophages (TAMs) to induce angiogenesis. In addition to promoting angiogenesis, TAMs may also activate other pathways to promote invasion and metastasis through secretion of various factors including IL-6, COX2, matrix metalloproteinase 9 (MMP9) and MMP14. How important it could be in exploring the relationship TIME and cancer progression? Please resume this point in the discussion.

6) Recently, Muller P and colleagues (Cancer Immunol Immunother. 2016) suggested that the analysis of the immune cell infiltration in matched primary and metastatic lesions from patients with NSCLC revealed lower CD8+:CD4+ T cell and CD8+:CD68+ T cell ratios in metastases (lung, brain, liver, distant lymph nodes, kidney and bone). Did you observe in your clinical case a tolerogenic effect and tumour-promoting microenvironment at the metastatic site? Please justify it.

Author Response

Responses to the comments from Reviewer #1

We thank the Reviewer #1 for helpful suggestions. Specific responses to the Reviewer’s comments follow below.

Comment 1;

The resolution of the figure 3F (Fluorescence in situ hybridization analysis of the ALK locus) needs to be improved. You might specify by arrows the exact point in which the ALK rearrangement occurs and the partner of ALK fusion (EML4 or others?).

Response to this comment:

We would like to thank the Reviewer for this comment. We agree with the Reviewer’s comment. The Figure 2F shows the split assay for the ALK gene to clinically detect ALK-fusion gene. The upstream and downstream of the ALK gene are labeled red and green, respectively. If the ALK gene is translocated, the signals appear separated. According to the Reviewer’s suggestion, we have added arrowheads, which indicate split pattern signals, into the revised Figure 2F. We have revised Figure Legend of Figure 2F. According to the Reviewer’s suggestion, we also tried to improve the resolution and replaced the Figure 2F with a revised picture.

We have incorporated the following sentences into the Figure Legend of Figure 2F. (Page 4, lines 122-126).

              Fluorescence in situ hybridization (FISH) analysis of the ALK locus using a break-apart probe strategy. The upstream and downstream of ALK gene were labeled red and green, respectively. Approximately 58% of tumor cells showed rearrangement at the ALK locus, as demonstrated by split red/green signals. Arrowheads indicate split pattern signals. ALK, anaplastic lymphoma kinase.

Comment 2;

I suggest to the authors to resume the materials and methods, especially the Fluorescent multiplex immunohistochemistry section.

Response to this comment:

We would like to thank the reviewer for this comment. We agree with the reviewer’s comment. According to the Reviewer’s suggestion, we have revised the Materials and Methods section in the revised manuscript (Page 11, lines 355-369).

Comment 3;

As in any case study, the present study has important limitations.

  • It involves only one patient, and thus further studies are needed to determine whether the principles discovered here apply to other patients.

Response to this comment:

We would like to thank the reviewer for this comment. We agree with the reviewer’s comment. We recognize our study has critical limitations. According to the Reviewer’s suggestion, we revised the Discussion section and have incorporated the following sentence into the revised Discussion section (Pages 9-10, lines 303-332).

              Incorporated sentences into the Discussion section: The present study has notable limitations. Only one patient is involved in this study, thus further studies are needed to determine whether the principles discovered here apply to other lung cancer patients.

Comment 4;

2) The interplay between treatment, somatic mutations, the immune system, and heterogeneous fates of the tumors cannot be untangled in this clinical case. For example, it is feasible that the ALK inhibitors treatment for this patient contributed to shaping the somatic mutations and the microenvironment of the tumors, but due to the descriptive nature of the study it was not possible to explore it. Please discuss this point.

Response to this comment:

We would like to thank the reviewer for this insightful comment. We agree with the reviewer’s comment. We consider that cancer therapy including ALK-inhibitors may change the TIME in primary and metastatic tumor lesions. In current study we did not address the impact of ALK-TKI on TIME (tissues after treatment). Thus, further studies evaluating TIME pre- and post-therapy are required to understand the complex interplay among cancer cells, host-immunity, and cancer therapy. According to the Reviewer’s suggestion, we have revised the Discussion section and have incorporated the following sentence into the Discussion section (Page 10, lines 316-319).

              Incorporated sentences into the Discussion section:  Cancer therapy including ALK-inhibitors may change the TIME in primary and metastatic tumors. In current study we did not address the impact of ALK-TKI on TIME. Thus, further studies evaluating TIME pre- and post-therapy are needed to understand the complex interplay among cancer cells, host-immunity, and cancer therapy.

Comment 5;

3)  Moreover, whole-exome sequencing and RNA-sequencing would have been important to deeply investigate the genetic, molecular, and cellular components that potentially underlie this differential growth. For example, I wonder if exists any evidence of gene-expression or somatic-alteration patterns (mutations, DNA amplification, and deep deletion) that differed between progressing (primary and distant metastases including mediastinal lymph-node, bilateral breasts, multiple bones, liver and brain) and regressing/stable tumors. Please justify this issue and explain why you didn’t perform NGS sequencing applied to this clinical case.

Response to this comment:

We would like to thank the reviewer for this insightful comment. We agree with the reviewer’s comment. Whole-exome sequencing and RNA-sequencing are important approaches to dissect the heterogeneity of complex biological systems (Maynard et al., Cell 2020;182, 1232–1251). These analyses of primary and metastatic lesions pre- and post-therapycould provide critical information to understand the complex interplay among cancer cells, host-immunity, and therapy. However, we could not these analyses in our patient due to the following reasons. The current patient showed rapid progression of tumors at diagnosis and her performance status was 2. In addition, the patient’s tissue samples obtained from biopsies were limited. Therefore, we first tested ALK-protein expression and epidermal growth factor receptor (EGFR) mutation status. Then we found that the tumors were positive for ALK-rearrangement but negative for EGFR mutation. Although next-generation sequencing (NGS) (Oncomine Dx Target Test multi-CDx system) and microsatellite instability (MSI) tests are clinically available now, however, these analyses generally need more time to get the results. Therefore, we prioritized tests for ALK-fusion gene and EGFR mutation to initiate immediately patient’ care. 

According to the Reviewer’s suggestion, we revised the Result section and have incorporated the following sentence into the Result section (Page 2, lines 86-87).

              Incorporated sentences into the Result section: The patient showed rapid progression of tumors and her performance tatus was 2.

We also recognize our study has critical limitations. To show the limitations of current study, we have also revised the Discussion section and have incorporated the following sentence (Page 10, lines 319-323). We have also included relevant references in the revised manuscript.

              Incorporated sentences into the Discussion section:

Whole-exome sequencing and RNA-sequencing are important approaches to dissect the heterogeneity of complex biological systems (Maynard et al., 2020, Cell 182, 1232–1251, Genome Biology (2016) 17:231, Science. 2016 Mar 25;351(6280):1463-9. Cancer Cell. 2018 May 14;33(5):853-861.e4.). Analyses of primary and metastatic tumors bywhole-exome sequencing and RNA-sequencing pre- and post-therapy could provide a crucial information to understand the complex interplay among cancer cells, host-immunity, and cancer therapy.

Comment 6;

4)  Tumors that are genetically more heterogeneous have less immune infiltrates

(Senbabaoğlu et al., Genome Biol. 2016) and less benefit from checkpointblockade

immunotherapies (McGranahan et al., Science. 2016). Thus, the exploration of the tumor-immune microenvironments (TIME) as a prognostic, diagnostic or predictive biomarker is an area of active research. In fact, TMB was predictive of clinical efficacy independent of PD-L1 expression, suggesting that both TMB and PD-L1 can be used as biomarkers for patient selection (Hellmann MD et al. Cancer Cell 2018). Recently, the isolation and culture of circulating tumor cells (CTCs) may be an alternative to tumor biopsies for the diagnosis of ALK rearrangement. Additionally, the effectiveness of ALK detection by FISH using cultured CTCs could be an interesting point to prove. What's your opinion about this? Please brief reply in the discussion.

Response to this comment:

We would like to thank the reviewer for this insightful comment. We agree with the reviewer’s comment. As the Reviewer pointed out, the predictive value of PD-L1 has been validated in immune checkpoint blockade therapies of NSCLC (N Engl J Med. 2017 Jun 22;376(25):2415-2426.). It has been shown that tumor-mutation burden (TMB) was predictive of clinical efficacy independent of PD-L1 expression, suggesting that both TMB and tumor PD-L1 can be used as biomarkers for patient selection (Cancer Cell. 2018 May 14;33(5):853-861.e4., N Engl J Med. 2017 Jun 22;376(25):2415-2426.). Circulating tumor cells (CTCs) are cancer cells that are shed from the primary or metastatic tumors into the bloodstream. The enumeration and characterization of CTCs provided a minimally invasive and, therefore, repeatable, method despite being present in extremely low numbers, enabling the sampling of tumor cells from peripheral blood and monitoring PD-L1 expression on tumor cells over time. In addition, EML4-ALK rearrangements have been reported to be found in circulating tumor cells (CTCs) (Cells 2020, 9, 1465). Accumulating evidence suggests that longitudinal evaluation of serial human specimens including tumor and blood samples during treatment (at pre-treatment, early-on-treatment, and progression time points) allows for deep analysis to unveil potential mechanisms of therapeutic resistance (Cell. 2020, 182, 1232–1251, Cell. 2017 Feb 9;168(4):707-723.). These approaches may contribute to develop treatment modalities to overcome intra-patient tumor heterogeneity.

According to the Reviewer’s suggestion, we have revised the Discussion section and have incorporated the following sentence into the Discussion section (Page 10, lines 323-333). We have also included relevant references in the revised manuscript.

Incorporated sentences into the Discussion section:

              Circulating tumor cells (CTCs) are cancer cells that are shed from the primary or metastatic tumors into the bloodstream (Int J Mol Sci. 2017 May 11;18(5):1035., Lancet Oncol 2015;16(2):177-86). The enumeration and characterization of CTCs provided a minimally invasive and, therefore, repeatable, method despite being present in extremely low numbers, enabling the sampling of tumor cells from peripheral blood and monitoring PD-L1 expression on tumor cells over time. In addition, EML4-ALK rearrangements have been reported to be found in CTCs (Cells 2020, 9, 1465). Thus, combining CTC and peripheral immune subset analyses may make it possible to longitudinally evaluate serial human specimens during treatment (at pre-treatment, early-on-treatment, and progression time points), which allow for deep analysis to unveil potential mechanisms of therapeutic resistance (Cell.2020, 182, 1232–1251, Cell. 2017 Feb 9;168(4):707-723.). These approaches may contribute to develop treatment modalities to overcome intra-patient tumor heterogeneity.

Comment 7;

5)  It is known as endothelial cells respond to vascular endothelial growth factor (VEGF) from various sources including tumour cells, natural killer (NK) cells and tumour-associated macrophages (TAMs) to induce angiogenesis. In addition to promoting angiogenesis, TAMs may also activate other pathways to promote invasion and metastasis through secretion of various factors including IL-6, COX2, matrix metalloproteinase 9 (MMP9) and MMP14. How important it could be in exploring the relationship TIME and cancer progression? Please resume this point in the discussion.

Response to this comment:

We would like to thank the reviewer for this insightful comment. We agree with the reviewer’s comment. We recognize our study has critical limitations. According to the Reviewer’s suggestion, we have revised the Discussion section and have incorporated the following sentence into the Discussion section (Page 10, lines 308-316). We have also included relevant references in the revised manuscript.

TAMs inhibit immuno-stimulatory signals and are implicated in the initiation and progression of the tumor, through the secretion of signaling molecules, such as vascular endothelial growth factor (VEGF), transforming growth factor beta (TGF-b), macrophage colony-stimulating factor (M-CSF), interleukins or chemokines (IL-10, IL-6, and CXCL-8) (J. Clin. Med. 2020, 9, 3226). Factors secreted by TAMs, such as TGF- b, VEGF, CCL8, COX-2, MMP9, and MMP2 also contribute to the metastatic properties of cancer cells. In addition, TAMs are responsible for resistant to conventional antitumor treatments, such as chemotherapy, radiotherapy, or immune checkpoint inhibitors. However, we did not investigate TAMs in TIME of primary and metastatic tumors, which is a limitation.

Comment 8;

6)  Recently, Muller P and colleagues (Cancer Immunol Immunother. 2016) suggested that the analysis of the immune cell infiltration in matched primary and metastatic lesions from patients with NSCLC revealed lower CD8+:CD4+ T cell and CD8+:CD68+ T cell ratios in metastases (lung, brain, liver, distant lymph nodes, kidney and bone). Did you observe in your clinical case a tolerogenic effect and tumour-promoting microenvironment at the metastatic site? Please justify it.

Response to this comment:

We would like to thank the reviewer for this insightful comment. We agree with the reviewer’s comment. We recognize our study has critical limitations. In current study, CD4 and CD68 expressions were not evaluated, thus we could not assess CD8/CD4 ratio and CD8/CD68 ratio in primary and metastatic tumors. According to the Reviewer’s suggestion, we have revised the Discussion section and have incorporated the following sentence into the Discussion section (Page 9, lines 267-273). We have also included relevant references in the revised manuscript.

Incorporated sentences into the Discussion section:

              In a recent study, Muller P and colleagues investigated the heterogeneity of immune cell infiltrates between primary NSCLC and corresponding metastases (Cancer Immunol Immunother (2016) 65:1–11). In this study, primary tumors and corresponding metastases from 34 NSCLC patients were extensively analyzed by immunohistochemistry for CD4, CD8, CD11c, CD68, CD163 and PD-L1. Interestingly, this study reported that the CD8/CD4 ratio and CD8/CD68 ratio were significantly reduced in metastatic tumors compared with the corresponding primary tumors, suggesting tolerogenic and tumor-promoting microenvironment at the metastatic site.

Reviewer 2 Report

The manuscript entitled "Heterogeneous tumor-immune microenvironments between primary and metastatic tumors in a patient with ALK rearrangement-positive large cell neuroendocrine carcinoma" highlighted the heterogeneous TIME between primary and metastatic lesions and provides new insights into the complex interplay between host-immunity and cancer cells in primary and metastatic lesions.

  • In the Methods section the Authors should clarify the ALK IHC clone adopted.
  • Do the Authors perform TTF1 IHC staining?

Author Response

Responses to the comments from Reviewer #2

Reviewer #2 (Reviewer Comments to the Authors)

The manuscript entitled "Heterogeneous tumor-immune microenvironments between primary and metastatic tumors in a patient with ALK rearrangement positive large cell neuroendocrine carcinoma" highlighted the heterogeneous TIME between primary and metastatic lesions and provides new insights into the complex interplay between host-immunity and cancer cells in primary and metastatic lesions.

Response to this comment:

We thank the Reviewer #2 for helpful suggestions. Specific responses to the Reviewer’s comments follow below.

Comment 1;

In the Methods section the Authors should clarify the ALK IHC clone adopted.

Response to this comment:

We would like to thank the Reviewer for this comment. We agree with the reviewer’s comment. Clone D5F3 (rabbit monoclonal antibody) was used for our ALK IHC. According to the Reviewer’s suggestion, we have added an antibody cline name used for ALK IHC. We have revised and incorporated the sentence into the Figure 2 legend of Result section (Page 4, line 122).

Comment 2;

Do the Authors perform TTF1 IHC staining?

Response to this comment:

We would like to thank the Reviewer for this comment. We stained TTF-1 and the tumor cells were positive for thyroid transcription factor-1 (TTF-1). According to the Reviewer’s suggestion, we have added this information in the Resultsection of revised manuscript (Page 3, line 105).
